# Pyruvate decarboxylase and thiamine biosynthetic genes are regulated differently by Pdc2 in *S. cerevisiae* and *C. glabrata*

**Christine L. Iosue, Julia M. Ugras**[ID]**, Yakendra Bajgain, Cory A. Dottor, Peyton L. Stauffer, Rachael A. Hopkins, Emma C. Lang, Dennis D. Wykoff**[ID]*

Department of Biology, Villanova University, Villanova, Pennsylvania, United States of America

* dennis.wykoff@villanova.edu

## Abstract

Understanding metabolism in the pathogen *Candida glabrata* is key to identifying new targets for antifungals. The thiamine biosynthetic (THI) pathway is partially defective in *C. glabrata*, but the transcription factor *Cg*Pdc2 upregulates some thiamine biosynthetic and transport genes. One of these genes encodes a recently evolved thiamine pyrophosphatase (*CgPMU3*) that is critical for accessing external thiamine. Here, we demonstrate that *Cg*Pdc2 primarily regulates THI genes. In *Saccharomyces cerevisiae*, Pdc2 regulates both THI and pyruvate decarboxylase (PDC) genes, with PDC proteins being a major thiamine sink. Deletion of *PDC2* is lethal in *S. cerevisiae* in standard growth conditions, but not in *C. glabrata*. We uncover cryptic *cis* elements in *C. glabrata* PDC promoters that still allow for regulation by *Sc*Pdc2, even when that regulation is not apparent in *C. glabrata*. *C. glabrata* lacks Thi2, and it is likely that inclusion of Thi2 into transcriptional regulation in *S. cerevisiae* allows for a more complex regulation pattern and regulation of THI and PDC genes. We present evidence that Pdc2 functions independent of Thi2 and Thi3 in both species. The C-terminal activation domain of Pdc2 is intrinsically disordered and critical for species differences. Truncation of the disordered domains leads to a gradual loss of activity. Through a series of cross species complementation assays of transcription, we suggest that there are multiple Pdc2-containing complexes, and *C. glabrata* appears to have the simplest requirement set for THI genes, except for *CgPMU3*. *CgPMU3* has different *cis* requirements, but still requires Pdc2 and Thi3 to be upregulated by thiamine starvation. We identify the minimal region sufficient for thiamine regulation in *CgTHI20*, *CgPMU3*, and *ScPDC5* promoters. Defining the *cis* and *trans* requirements for THI promoters should lead to an understanding of how to interrupt their upregulation and provide targets in metabolism for antifungals.

## Introduction

Yeast pathogens have become increasingly dangerous in recent decades due to the use of wide-range antibiotics, surgeries, and catheters, while treatment of fungal infections has led to the appearance of antifungal resistance in yeast. The yeast species *Candida glabrata* is the second

**Data Availability Statement:** All relevant data are within the paper and its Supporting Information files.

**Funding:** This work was funded by National Science Foundation grant (MCB 1921632 to D.D. W.), the Dennis M. Cook Endowed Gregor Mendel Chair in Genetics Endowment, the Villanova College of Liberal Arts and Sciences, and the Villanova Department of Biology. The funders had no role in study design, data collection and analysis, decision to publish, or preparation of the manuscript.

**Competing interests:** The authors have declared that no competing interests exist.

most common source of candidiasis, and acquires antifungal resistance relatively easily [1]. *C. glabrata* is more closely related to the relatively nonpathogenic yeast *Saccharomyces cerevisiae* as opposed to other *Candida* pathogens [2, 3]. *C. glabrata* and *Saccharomyces cerevisiae* are both *Saccharomycetaceae* species that diverged from other yeast species after a whole genome duplication event in a common ancestor [4]. We have focused on thiamine starvation in these two species because they are appealing model systems to compare evolutionary conservation of function, and it is a case where there are significant differences between the two species [5–8]. Additionally, by understanding the thiamine starvation pathway (THI pathway), we may be able to identify druggable targets in *C. glabrata*.

Thiamine is required for all life forms as it is required for glucose catabolism, and thiamine pyrophosphate (TPP) is the essential, functional cofactor for amino acid and carbohydrate metabolism [9]. *S. cerevisiae* can synthesize thiamine *de novo* and utilize extracellular thiamine sources to synthesize TPP. In *S. cerevisiae*, two transcription factors, Thi2 and Pdc2, and one regulatory factor, Thi3, modulate the activation of THI genes in response to thiamine starvation. Thi3 is related to the major TPP binding proteins, pyruvate decarboxylases, and when TPP is abundant in cells, TPP occupies a pseudo active site in Thi3p, destabilizing the transcriptional complex of Thi3, Thi2, and Pdc2 [10]. Under thiamine starvation conditions, TPP is dissociated from Thi3, and Thi3 interacts with the C-terminal domain of Pdc2. Pdc2 can then recruit transcription factors efficiently, and the transcription factor complex drives the transcription of THI genes [11]. While this model has not been fully supported, the thought is that due to this dynamic interaction between TPP and Thi3, the Pdc2 interaction with THI promoters is sensitive to intracellular TPP concentration with a Thi2/Thi3/Pdc2 complex forming at THI promoters [12].

*C. glabrata* lacks the transcriptional coactivator Thi2 and cannot synthesize the pyrimidine precursor for thiamine, making *C. glabrata* auxotrophic for thiamine. However, *C. glabrata* upregulates a small set of genes to acquire thiamine and thiamine precursors during thiamine starvation [6]. Overexpression of *CgTHI3* can partially compensate for the loss of *THI2* in *S. cerevisiae*, suggesting that Pdc2 can be activated independent of the Thi2 transcription factor [6]. Despite the loss of some THI genes, such as *THI2*, *C. glabrata* uniquely acquired the gene *CgPMU3*, which is upregulated during thiamine starvation and not found in any other species [8]. *CgPMU3* is a newly evolved gene that encodes a TPP phosphatase, which is needed to utilize external TPP, and serves as a functional analog to *Sc*Pho3. Consequently, deleting *CgPMU3* leads to a growth defect in an environment with TPP as the primary thiamine source, such as human serum [8, 13]. Thus, there are significant differences between the two species in what is upregulated in response to thiamine starvation, and what factors are required for that upregulation.

While required for upregulation of thiamine biosynthesis genes, Pdc2 is essential for expression of pyruvate decarboxylase genes (PDC genes), including *ScPDC1* and *ScPDC5* in *S. cerevisiae* [14]. Deletion of *ScPDC2* is lethal in medium with high glucose concentrations, and it is thought this is because the PDC genes are not sufficiently expressed to metabolize glucose [15]. Overexpression of *ScPDC1* suppresses the growth defect of the *Scpdc2*Δ strain [12]. Pdc2 also interacts with the promoter of *PDC5* (*PDC5*pr), which encodes a pyruvate decarboxylase isoform synthesized under thiamine starvation conditions [12, 14]. *PDC5*pr is utilized for investigating Pdc2 binding because it is dependent on Pdc2, but independent of Thi2 and Thi3, making it a simpler promoter to distinguish the binding site for Pdc2. Nosaka et al. identified an upstream region of *PDC5*pr between -416 and -346 nucleotides that is necessary for *ScPDC5* expression, and this region shares sequence similarity with a binding site in THI promoters that we identified in *C. glabrata* [7, 12]. Because deletion of *ScPDC2* is lethal, few genome-wide studies have examined Pdc2 properties in *S. cerevisiae*.

For this reason, and because *Sc*Pdc2 appears to regulate different kinds of genes (THI and PDC genes) and have different requirements for cofactors, we examined the properties of Pdc2 in both species in detail.

We undertook this work to understand the roles of Pdc2 in both species and to understand the different requirements of Pdc2-dependent promoters. We demonstrate: 1) the Pdc2 activation domain is critical for essentiality in *S. cerevisiae*. 2) The PDC and THI promoters have different requirements for transcriptional regulators with Pdc2. 3) *S. cerevisiae* and *C. glabrata* have different requirements at Pdc2 dependent promoters which correlates with the loss of Thi2. 4) The activation domain of Pdc2 exhibits characteristics consistent with intrinsically disordered regions (IDRs) [16], but loss of parts of these IDRs does not affect promoters differentially. 5) Finally, we identify the minimal *cis* regions for different classes of promoters that are sufficient to confer thiamine starvation upregulation to a basal promoter.

## Methods

### Strains

Strains used in this study are listed in **S1 Table**. Experiments were performed in *S. cerevisiae* wild-type and *C. glabrata* wild-type, as well as strains in which the thiamine pathway regulators were deleted: *Scpdc2Δ*, *Scthi3Δ*, *Scthi2Δ*, *Cgpdc2Δ*, and *Cgthi3Δ* [6, 7, 17–19].

To swap Pdc2 proteins between *S. cerevisiae* and *C. glabrata*, *PDC2* was first deleted with *URA3*, which replaced the open reading frame (ORF) via homologous recombination [20]. Deletions were verified by gain of the *URA3* marker as well as by PCR to confirm loss of the gene and positivity for flanking PCR regions. To construct a *Scpdc2*::*URA3* strain capable of growth in glucose medium, *ScPDC1* was overexpressed under the control of the *ScADH1* promoter on a *LEU2*[+] plasmid (pRS315) [7] in *S. cerevisiae* wild-type before deletion of *PDC2*. We chose to perform all assays with standard growth conditions because the *Scpdc2* strain grows slowly in ethanol/glycerol containing medium. The open reading frame of *PDC2* from both species was then amplified and transformed into the *pdc2*::*URA3* strains to precisely replace the ORFs. Strains were confirmed by loss of the selectable *URA3* marker as well as by PCR. (All primer sequences listed in **S2 Table**).

To measure expression of *C. glabrata* THI promoters when *Cg*Pdc2 was truncated, the promoters were fused to yellow fluorescent protein (YFP) and incorporated into the genome, rather than introduced on a plasmid, to minimize noise from a plasmid reporter. The promoter-YFP was amplified to precisely replace the *CgURA3* ORF in *C. glabrata* wild-type using homologous recombination. Strains were verified by loss of *URA3*, by PCR, and by presence of fluorescence during thiamine starvation, indicating expression of the promoter. *CgPDC2* was then deleted in these fluorescent strains using *URA3* as a selectable marker. Strains were verified by gain of *URA3*, by confirmatory PCR, and by loss of fluorescence during thiamine starvation, indicating *PDC2* was deleted. *URA3* was then replaced by *CgPDC2*, either full-length or truncated in 40 amino acid increments from the C-terminus (amplified from the plasmids described below), and strains were again confirmed by loss of the *URA3* marker and by PCR.

### Plasmids

To assess complementation of the *Scpdc2Δ* and *Cgpdc2Δ* strains, *ScPDC2*, *CgPDC2*, or fusions of the DNA binding domain and activation domain of these two proteins, were cloned by homologous recombination into a *HIS3*[+] plasmid (pRS313) containing the *CgPDC2* promoter [20, 21]. Plasmids were verified by PCR and by complementation in the native species' deletion strain.

To assay induction of pyruvate decarboxylase (PDC) and thiamine (THI) pathway genes, we constructed plasmids where the promoter region of each gene is fused to YFP, allowing expression to be measured via fluorescence. The promoters (1 Kbp or 2 Kbp upstream of the start codon) were amplified by PCR and cloned by homologous recombination into a *HIS3*⁺ plasmid (pRS313) containing

YFP in a wild-type strain [20–22]. Plasmids were verified by PCR and fluorescence in a wild-type strain in the appropriate growth conditions.

To assess the ability of small regions from THI promoters to confer thiamine regulation to an unregulated promoter *CgPMU1*, the THI promoter regions and *CgPMU1* promoter were amplified by PCR to have overlapping sequences and all three PCR products were cloned by homologous recombination into a *HIS3*⁺ plasmid (pRS313) containing YFP. THI promoter regions were inserted into the *CgPMU1* promoter approximately the same distance upstream of the start codon as in the native THI promoter (see schematic in Fig 8A). Plasmids were confirmed by PCR and whole plasmid sequencing (Plasmidsaurus).

To assay induction of THI promoters when *CgPDC2* is truncated, truncated regions of *CgPDC2*, with the *CgPDC2* promoter and terminator, were amplified by PCR and cloned by homologous recombination into a *HIS3*⁺ plasmid (pRS313). Plasmids were confirmed by PCR. (All primers used are listed in **S2 Table**).

## Flow cytometry

To assay induction of pyruvate decarboxylase (PDC) and THI pathway promoters, fluorescence of cells containing plasmids with promoters fused to YFP was quantified by flow cytometry. Cells were grown at 30˚C overnight in thiamine replete SD medium lacking histidine (Sunrise Science, CA). Cells were harvested by centrifugation, washed three times with sterile water, inoculated into thiamine replete (0.4 mg/L) and starvation (no thiamine added) conditions in SD medium lacking histidine, and grown at 30˚C overnight (~18 hours). Mean fluorescence (in arbitrary units, a.u.) of each strain was measured using a flow cytometer with a 533/30 filter set (Accuri C6 Plus, BD Biosciences). In almost all cases, background fluorescence was less than 10,000 a.u.; however, there is variability of fluorescence based on precise growth conditions and we included positive and negative controls in each experiment.

## Results

### Both *Sc*Pdc2 and *Cg*Pdc2 regulate thiamine starvation regulated genes, but *Sc*Pdc2 is critical for pyruvate decarboxylase gene expression, making its loss lethal in standard growth conditions

Deletion of *ScPDC2* is lethal to cells grown in high glucose conditions, whereas deletion of *CgPDC2* does not result in an obvious growth defect, suggesting that the transcription factor has different specificities in the two yeast species. To understand the role of Pdc2 on the growth properties of *S. cerevisiae* and *C. glabrata*, we deleted *PDC2* in both species. As Pdc2 is not essential for growth in *C. glabrata*, we were able to delete *PDC2* normally. In *S. cerevisiae*, Pdc2 is essential for growth in 2% glucose conditions, and so we covered the *Scpdc2* with *ScPDC2* on a *URA3*⁺ containing plasmid. This allowed us to counter select against the plasmid with 5-FOA and transform in plasmids containing *ScPDC2* or *CgPDC2* on a *HIS3*⁺ containing plasmid to determine which can complement the growth defect of the *Scpdc2* strain. We confirmed previous work that demonstrated that *ScPDC2* is essential in standard glucose rich medium. Additionally, we determined that *CgPDC2* was unable to rescue the lethal phenotype (**Fig 1A**). Finally, we confirmed that overexpression of *ScPDC1* (under the control of the

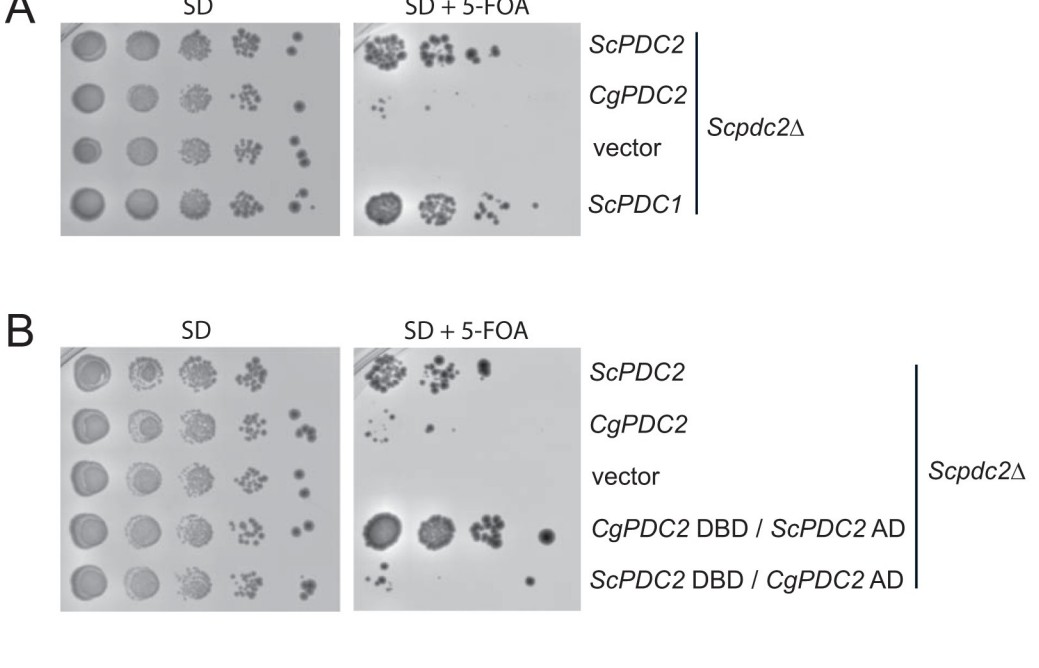

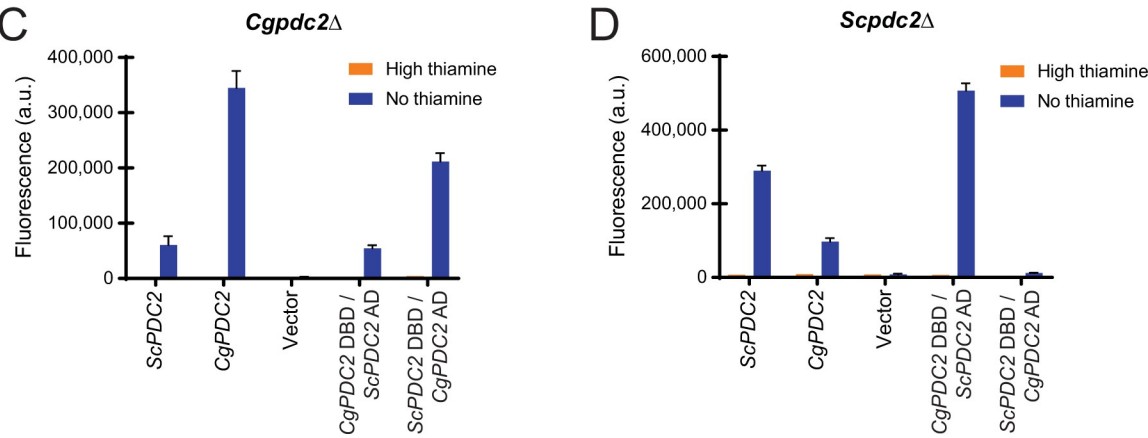

**Fig 1. Pdc2 is critical for pyruvate decarboxylase gene expression in *S. cerevisiae*, but regulates thiamine starvation regulated genes in both *S. cerevisiae* and *C. glabrata*.** The activation domains (AD) dictate specificity. (A-B) Growth of a *Scpdc2Δ* strain with *ScPDC2*, *CgPDC2*, *ScADH1pr*-Sc*PDC1*, or fusions of the DNA binding domain (DBD) and activation domain (AD) of both species' Pdc2 proteins introduced on a plasmid. (C) Expression of *CgTHI4* promoter fused to YFP in a *Cgpdc2Δ* strain and (D) *ScTHI4* promoter fused to YFP in a *Scpdc2Δ* strain with *PDC2* introduced on a plasmid. The fluorescence data presented is the mean and standard deviation of three independently grown samples.

*ScADH1* promoter) was able to rescue the lethality of the *Scpdc2Δ* (**Fig 1A**), which had been previously reported [12].

Given that the DNA binding domain (DBD) in the N-terminus (1–485 aa in *C. glabrata* and 1–494 aa in *S. cerevisiae*) shares 78% identity (87% similarity) to one another and the C-terminal activation domains (AD) have 22% identity (42% similarity), we hypothesized that the DBDs were likely interchangeable and the AD is likely the critical determinant for gene regulation specificity. By placing fusions of *Sc*Pdc2 DBD–*Cg*Pdc2 AD and *Cg*Pdc2 DBD–*Sc*Pdc2 AD into the *Scpdc2Δ* strain, we were able to determine that the *Sc*Pdc2 AD is important for viability by its rescue of lethality in standard medium (**Fig 1B**). Whereas the *S. cerevisiae*

and *C. glabrata* Pdc2 proteins have different functions for viability, both regulate thiamine starvation regulated genes as demonstrated with promoter-YFP fusion expression (**Fig 1C**—*Cgpdc2* and **1D**—*Scpdc2*); swapping Pdc2 in each species allows for the transcriptional induction of *THI4*, at least partially.

To determine whether the activation domains in each species dictate the ability to induce THI genes, we examined the Pdc2 fusions from **Fig 1B**. In *C. glabrata*, the AD is the primary determinant to induce transcription, suggesting the AD interacts with general transcriptional machinery to drive the upregulation. In *S. cerevisiae*, the story is different. The AD of *S. cerevisiae* seems to be the important determinant (compare ScPdc2 to CgPdc2 DBD/ScPdc2 AD), but the opposite fusion seems to remove the ability of Pdc2 to function in *S. cerevisiae* (compare CgPdc2 to ScPdc2 DBD/CgPdc2 AD). Because this same plasmid functions in *C. glabrata* (**Fig 1C**), and the entire *C. glabrata* Pdc2 protein can complement the *Scpdc2* (**Fig 1D**), this suggests that there is antagonism between the *S. cerevisiae* DBD and the *C. glabrata* AD, but only in *S. cerevisiae*. Given that *S. cerevisiae* also has Thi2 (and *ScTHI4*pr is dependent on Thi2), it suggests that this antagonism might disrupt the interactions in the Pdc2/Thi3/Thi2 complex. This is not surprising because the *Sc*Pdc2 AD region (407–925 aa) has been shown to interact with Thi3 in two hybrid assays [11]. It is worth noting that the *Sc*Pdc2 DBD/*Cg*Pdc2 AD fusion still functions in *C. glabrata* which suggests it can still interact with Thi3 at least in *C. glabrata*. In sum, we believe the presence/absence of Thi2 leads to this contradictory data.

## *PDC2* is more important for regulating pyruvate decarboxylase (PDC) genes in *S. cerevisiae* relative to *C. glabrata*

Given that the *Scpdc2*Δ strain is inviable in glucose containing medium, but the *Cgpdc2*Δ is viable, we explored the regulation of PDC genes in both species. *S. cerevisiae* contains three PDC genes. *ScPDC1* is thought to be the major PDC isoform in standard medium, *ScPDC5* is thought to be a specialized thiamine repressible PDC, and of note both open reading frames are very similar to one another (88% identity) [14, 23, 24]. *ScPDC6* (84% identity with *ScPDC1*) is thought to be important for growth on non-fermentable carbons sources [25]. *ScTHI3* is 52% identical to *ScPDC1*, but it is not catalytically active (it is a regulator of Pdc2). Thus, we choose to focus on the two important PDCs in standard medium (*ScPDC1* and *ScPDC5*). *C. glabrata* has only two copies of PDC genes (*CgPDC1—CAGL0M07920g* and *CgPDC5—CAGL0G02937g*) and *CgTHI3*. There are other weakly related proteins that are likely carboxylases with substrate specificity different from pyruvate.

To understand the regulation of the 4 genes (*ScPDC1*, *ScPDC5*, *CgPDC1*, and *CgPDC5*), we cloned 1 kb of each promoter upstream of YFP in a plasmid and examined YFP expression in wild-type, *pdc2*Δ, *thi3*Δ, and *Scthi2*Δ (**Fig 2**). We examined expression in both high thiamine and thiamine starvation conditions (no thiamine), because it was known that *ScPDC5* is regulated by thiamine starvation [26]. We observe in *S. cerevisiae* that *ScPDC5*pr is expressed at a low level in high thiamine conditions and is induced >50-fold during starvation. We note that both *ScPDC1*pr and *ScPDC5*pr are dependent on *ScPDC2*, explaining why loss of *PDC2* is so detrimental to a cell in standard glucose conditions (**Fig 2A**). Interestingly, Hohmann's group demonstrated that loss of *ScPDC1* is not lethal, and leads to an upregulation of *ScPDC5* even in high thiamine conditions [23, 26, 27]. We confirmed that in our assay, *ScPDC5*pr-YFP expression is increased in a *Scpdc1*Δ strain relative to wild-type even in high thiamine conditions (**S1 Fig**). This indicates that there are feedback mechanisms that impact PDC expression independent of thiamine status, and that those feedback mechanisms likely are a response to lack of pyruvate decarboxylase activity the cell. It is unclear whether PDC expression can be independent of *Sc*Pdc2.

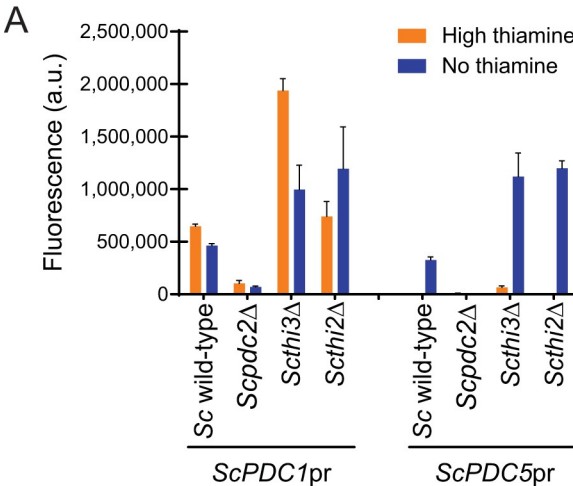

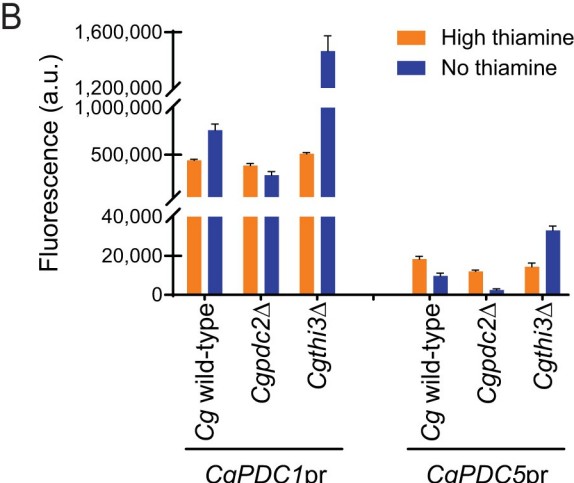

**Fig 2. Pdc2 is more important for regulation of pyruvate decarboxylase (PDC) genes in *S. cerevisiae* than *C. glabrata*.** Expression of *PDC1* and *PDC5* promoters fused to YFP in high thiamine and thiamine starvation (no thiamine) in (A) *S. cerevisiae* and (B) *C. glabrata* strains where the thiamine regulatory factors were deleted. The data presented is the mean and standard deviation of three independently grown samples.

Finally, we note a consistent elevation of expression of both *S. cerevisiae* genes in the absence of *THI3* or *THI2*, indicating that expression does not require these factors. This suggests that *Sc*Pdc2 has roles in transcription independent of thiamine starvation. Given that data has indicated that *Sc*Pdc2 is in a complex with Thi3 and Thi2 during thiamine starvation (and even in high thiamine to a lesser extent) [11, 12], ***our data suggest that Thi3 and Thi2 inhibit Pdc2 at PDC promoters***. A simple explanation would be that the Pdc2/Thi3/Thi2 complex is unable to bind PDC promoters, and Pdc2 alone, or with other factors, binds PDC promoters.

In *C. glabrata*, we do not observe the same complexity (**Fig 2B**). Loss of *CgPDC2* impacts expression of *CgPDC5* but only reduces *CgPDC1* expression by two-thirds, likely explaining why deletion of *CgPDC2* is not lethal–there is still significant expression of *CgPDC1*. We believe that factors other than Pdc2 might regulate these promoters, and the role of Pdc2 is

diminished. Loss of *THI3* does alter expression of *CgPDC1* and *CgPDC5* expression during thiamine starvation like *S. cerevisiae*, suggesting that Pdc2 could have a role in regulating these promoters, just in a diminished capacity. From these data, it is clear *PDC2* is not as important for expression of PDCs, and it would suggest that deletion of *PDC2* is not lethal in *C. glabrata* because its deletion does not severely reduce PDC expression, but in *S. cerevisiae* Pdc2 is crucial for PDC regulation.

## Regulation of THI genes in both species

Given the different requirements for transcription of PDC genes between the species, the question arose: what are the requirements of THI gene promoters in the two species? To examine this, we chose three types of THI genes: the major thiamine uptake transporter *THI10*, two thiamine biosynthetic genes *THI4* and *THI20*, and the TPPase *CgPMU3* [8]. We did not study *ScPHO3* because deletion of *ScPDC2* did not have a large impact on its regulation [8, 28]. The choice of these genes spans both biosynthetic genes and acquisition genes. We cloned 1 kb of these promoters fused to YFP into a plasmid and examined expression. *C. glabrata* once again is a simpler case. All classes of THI gene promoters are highly dependent on *CgPDC2* and *CgTHI3* (**Fig 3A**).

To determine whether these *C. glabrata* promoters functioned similarly in *S. cerevisiae*, we moved the promoters to *S. cerevisiae* strains and assayed expression (**Fig 3B**). Expression of *CgTHI4* and *CgPMU3* behave as if they are standard THI promoters in *S. cerevisiae*, albeit at a much lower level. Even though these promoters function without Thi2 in *C. glabrata*, they now are dependent on the Pdc2/Thi3/Thi2 complex present in *S. cerevisiae*. Expression of *CgTHI10* and *CgTHI20* is lost in *S. cerevisiae*, underlying that while all *C. glabrata* THI promoters appear simple in their requirements, there is more complexity at the promoter that we do not understand; the promoters behave identically in *C. glabrata* but not the same in *S. cerevisiae*. Interestingly, *CgPMU3* has a different *cis* region from all of the other *C. glabrata* THI promoters, raising the question of how a promoter that does not have the same *cis* architecture as other THI promoters is still responsive to the same *trans* environment in *S. cerevisiae* [7]. *CgPMU3* is a recently evolved phosphatase gene that cleaves thiamine pyrophosphate (TPP), is required for *C. glabrata* to survive when TPP is the only thiamine source, and is regulated by the same transcription factors as other THI genes. Because *CgPMU3* promoter does not use a *cis* element that is common to other THI promoters, we believe these data can begin to untangle the *cis* architectures required for regulation by thiamine status. This is because *CgPMU3* evolved recently, whereas all of the other THI genes have long common evolutionary histories and have likely experienced similar selection pressures over evolutionary time.

In *S. cerevisiae*, each class of promoters has slightly different requirements (**Fig 4**). *ScTHI10*, *ScTHI4*, and *ScTHI20* are dependent on *ScPDC2* and *ScTHI3*. *ScTHI2* is required for expression of *ScTHI20*, but not the *ScTHI10* transporter gene, and only partially for the biosynthetic gene *ScTHI4*. These data are consistent with previous work that indicates Thi2 is necessary for the expression of some, but not all THI genes in *S. cerevisiae* [10, 12, 14]. While qualitatively we observe that *S. cerevisiae* promoters often function in *C. glabrata*, this is true for *ScTHI10* and *ScTHI4*, both of which do not absolutely require Thi2 –i.e. these promoters do not need Thi2 and function fine with Pdc2 and Thi3. Interestingly, these *S. cerevisiae* THI promoters that do not require Thi2 are expressed at approximately the same level in either species. A simple explanation for this could be that the simple *Cg*Pdc2/*Cg*Thi3 complex effectively binds *cis* elements from either species, but the lack of Thi2 in *C. glabrata* prevents expression of some THI genes (like *ScTHI20*), and likewise, the *C. glabrata* promoters are expressed much less

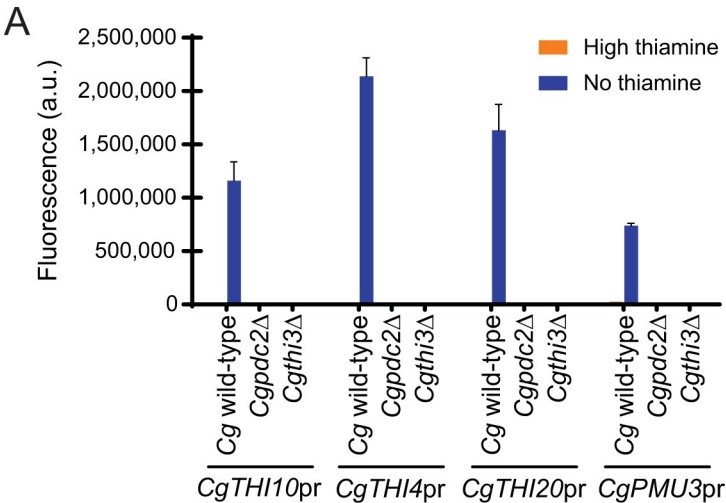

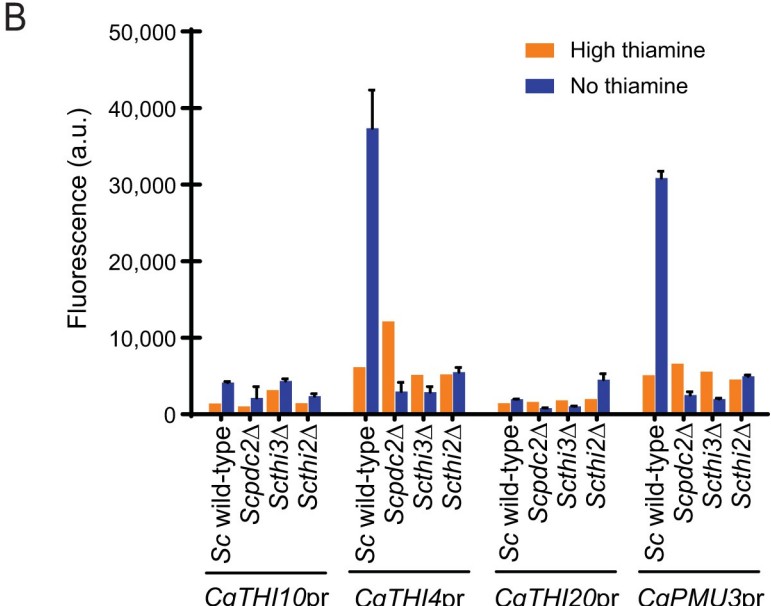

**Fig 3. *C. glabrata* THI promoters.** Expression of thiamine regulated promoters fused to YFP in high thiamine and thiamine starvation (no thiamine) in strains where the thiamine regulatory factors were deleted. *C. glabrata* promoters in (A) *C. glabrata* strains and (B) *S. cerevisiae* strains. The data presented is the mean and standard deviation of three independently grown samples.

efficiently because of Thi2 interference, and potentially, the sequestering of *Sc*Pdc2 binding to PDC promoters.

To dissect the *trans* effects of Pdc2 more thoroughly, we precisely replaced the open reading frames of *PDC2* in each species, allowing us to determine if having the appropriate species' version of Pdc2 allows for a restoration of expression. Here, we find that *C. glabrata* THI promoter-YFP plasmids require *CgPDC2* to fully induce expression, and swapping the species' Pdc2 eliminates most expression (**Fig 5A**). This indicates that in *C. glabrata* the THI promoters require some function of *Cg*Pdc2 that *Sc*Pdc2 cannot provide. Additionally, it suggests that just having *Cg*Pdc2 in *S. cerevisiae* is not sufficient for expression of *C. glabrata* promoters.

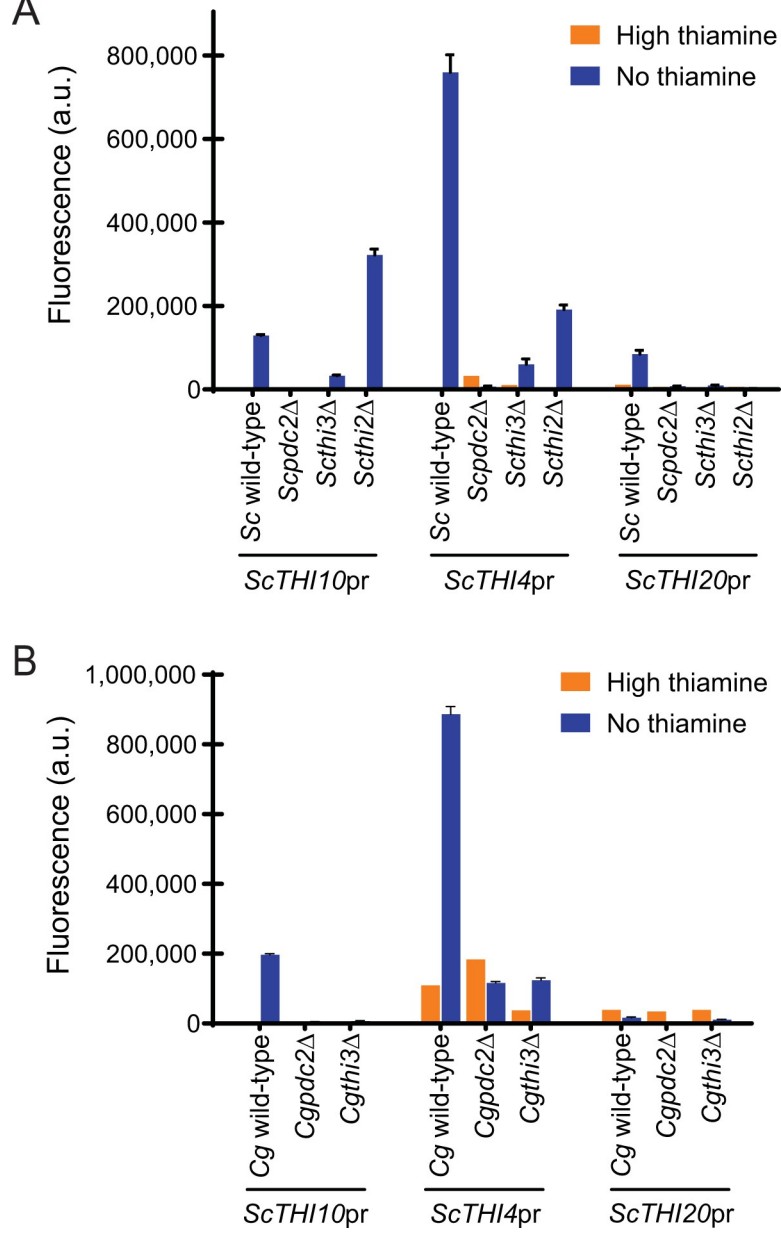

**Fig 4. *S. cerevisiae* promoters.** Expression of thiamine regulated promoters fused to YFP in high thiamine and thiamine starvation (no thiamine) in strains where the thiamine regulatory factors were deleted. *S. cerevisiae* promoters in (A) *S. cerevisiae* strains and (B) *C. glabrata* strains. The data presented is the mean and standard deviation of three independently grown samples.

Alternatively, *S. cerevisiae* promoters function in *S. cerevisiae*, and these promoters are only somewhat sensitive to which version of Pdc2 is present (**Fig 5B**). This supports the previous arguments that *S. cerevisiae* promoters are able to function in *S. cerevisiae* even when *Cg*Pdc2 is the transcription factor. However, when expressed in *C. glabrata*, the *S. cerevisiae* THI promoters function best with the *trans C. glabrata* factors, as moving *Sc*Pdc2 into *C. glabrata* attenuates expression. This, not surprisingly, suggests that each species has optimized how the protein-protein interactions occur, and that Pdc2 is core to those protein-protein interactions.

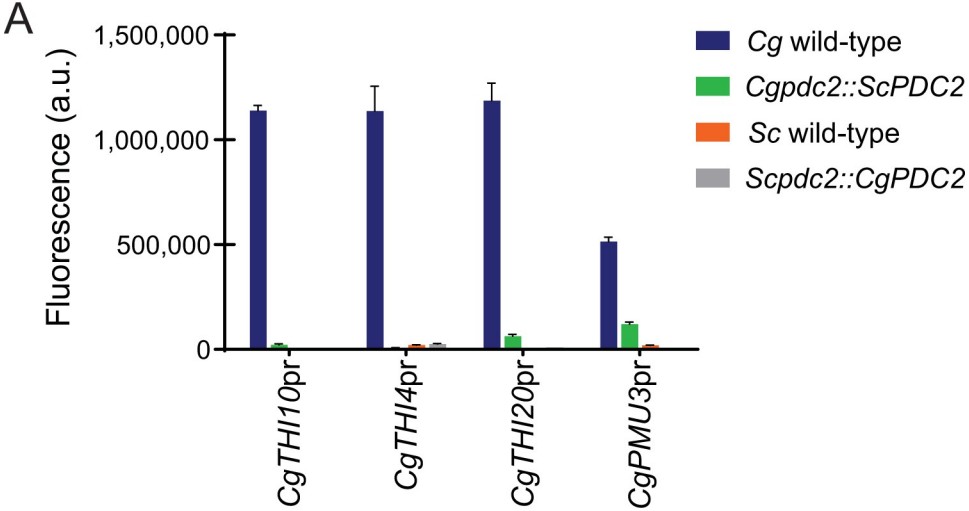

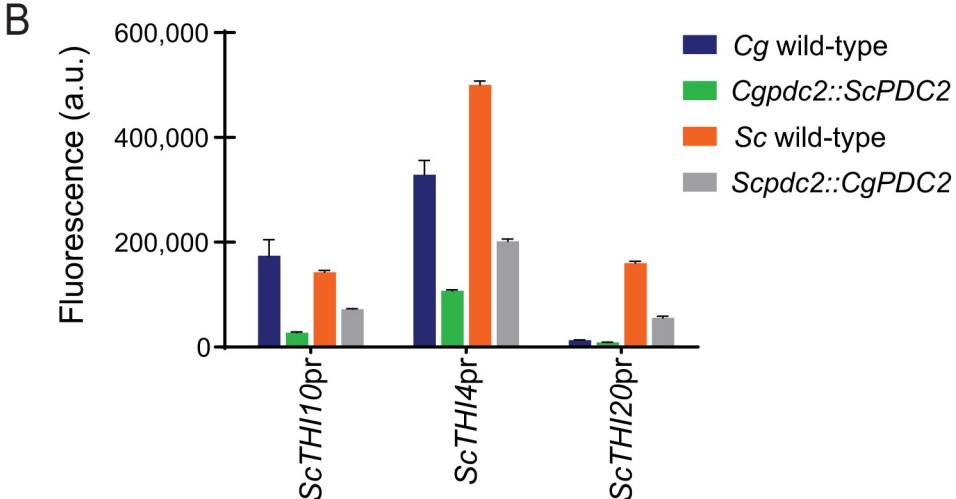

**Fig 5. *C. glabrata* PDC2 is necessary but not sufficient for the expression of *C. glabrata* THI promoters, while *S. cerevisiae* THI promoters can be regulated by *PDC2* from either species.** Expression of (A) *C. glabrata* and (B) *S. cerevisiae* THI promoters fused to YFP in high thiamine and thiamine starvation (no thiamine) in strains where the *PDC2* open reading frame was precisely replaced with *PDC2* from the opposite species. The data presented is the mean and standard deviation of three independently grown samples.

However, we cannot formally exclude the possibility that precise replacement of the open reading frame does not alter other factors, such as the stability of each protein in the other species.

## PDC1 and PDC5 promoters

The biggest surprise was with *CgPDC1* and *CgPDC5*, which appeared only subtly regulated by Pdc2 and Thi3 in *C. glabrata*, but now are highly induced during thiamine starvation in *S. cerevisiae* when *ScTHI2* and *ScTHI3* are deleted (**Fig 6A**). This effect is prominent with the *CgPDC1* promoter. This suggests that *CgPDC1* might have the remnants of *ScPDC1* regulation, but in *C. glabrata* other factors are acting on PDC promoters masking the role of *Cg*Pdc2. This

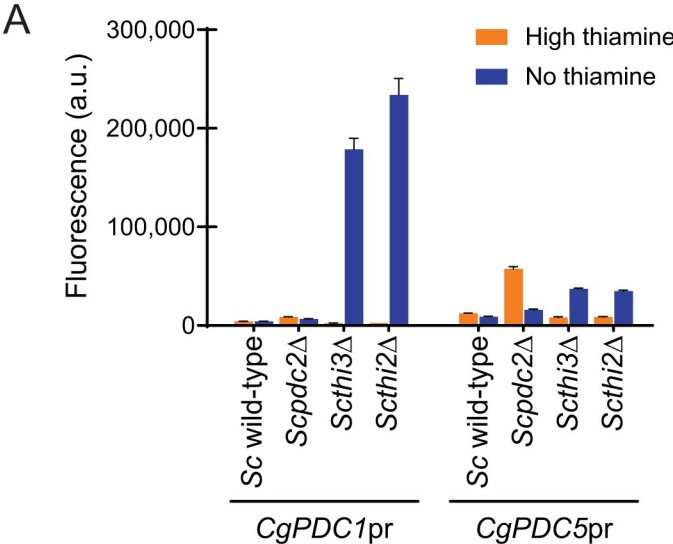

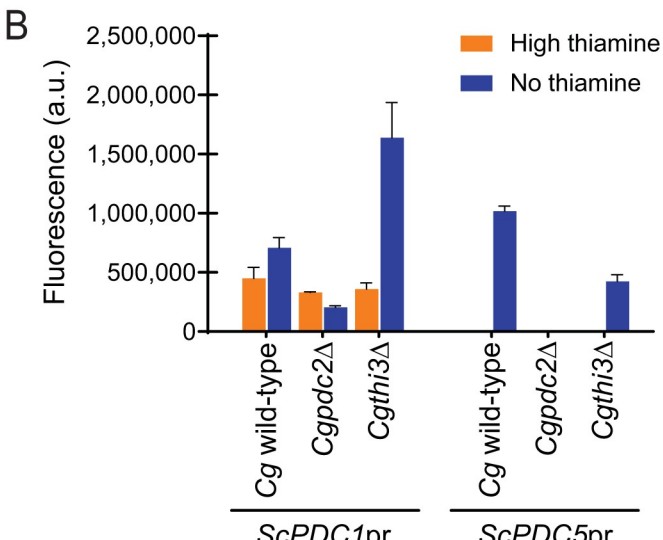

**Fig 6. PDC promoters in both species.** Expression of *PDC1* and *PDC5* promoters fused to YFP in high thiamine and thiamine starvation (no thiamine) in strains where the thiamine regulatory factors were deleted. (A) *C. glabrata* PDC promoters in *S. cerevisiae* strains. (B) *S. cerevisiae* PDC promoters in *C. glabrata* strains. The data presented is the mean and standard deviation of three independently grown samples.

is not surprising given it seems likely that the PDC promoters have complex regulation independent of thiamine as pyruvate decarboxylase activity is critical for glucose metabolism. Surprisingly, wild-type *S. cerevisiae* has low expression of *CgPDC1* and *CgPDC5* and no apparent increase in expression in response to thiamine starvation. It is worth noting that if such complexities are present, we are unable to disentangle them with these experiments. In *S. cerevisiae* wild-type, the native *ScPDC1* is being expressed to a high-level, potentially suppressing expression of *CgPDC1*, and in the *Scpdc2Δ* strain, where *ScPDC1* is expressed from an ectopic promoter (*ScADH1*pr), a similar situation is likely happening. In both cases, it seems likely that

*CgPDC1* is not expressed in *S. cerevisiae* because there are high levels of *Sc*Pdc1 in the cell. These results are actually not surprising given that Hohmann's group has demonstrated feedback mechanisms on *ScPDC5* in the absence of *ScPDC1* [23, 27]. The key result here is that loss of Thi3/Thi2 appears to release the constraint on Pdc2, allowing for high expression of *CgPDC1*. This indicates that the *CgPDC1* promoter likely binds ScPdc2 without Thi2/Thi3. A similar process appears to happen with *CgPDC5*, but to a lesser extent.

When the *ScPDC1*pr-YFP plasmid is introduced into *C. glabrata*, it behaves the same as *CgPDC1* in *C. glabrata* (compare **Figs 2B–6B**). This may indicate that both promoters have the ability to bind Pdc2 through the same *cis* elements and the *trans* milieu of *C. glabrata* dictates its expression. The converse is less obvious, but likely impacted by pyruvate decarboxylase activity feedback and the potential for *CgPDC1* to be regulated by additional factors as we see in *C. glabrata* (compare **Figs 2A–6A**). When we introduce *ScPDC5*pr-YFP into *C. glabrata*, we see it is completely dependent on *CgPDC2* and only partially dependent on *CgTHI3* (**Fig 6B**). *CgPDC5*pr-YFP is expressed at a low level in *S. cerevisiae*, but does have similarities to its expression in *C. glabrata* (**Fig 6A**). We believe these data suggest that the *trans* environment is more important for dictating expression of the PDC genes in both species, but the *cis* elements are still required. We believe that PDC gene regulation warrants future studies to tease apart the requirements further and are beyond the scope of this study.

All of the data, so far, present a surprisingly complex set of transcriptional regimes. In *S. cerevisiae* there are cases where it appears that Pdc2 may act independent of the Thi2 and Thi3 regulators, where Pdc2 may require Thi3 but not Thi2, and where all three proteins likely form a Pdc2/Thi3/Thi2 complex. In *C. glabrata* the diversity of different complexes appears reduced, with PDC genes possibly only being partially dependent on Pdc2 and THI genes absolutely requiring both Pdc2 and Thi3 in a complex.

### Both the DBD and AD of Pdc2 is required for the expression of *C. glabrata* genes and sequential loss of the intrinsically disordered regions (IDRs) in the AD impacts promoters in the same way

*CgPMU3* having a different critical *cis* element relative to other promoters suggests that the architecture of the *CgPMU3* promoter is different from other THI promoters, even though *CgPMU3* uses both Pdc2 and Thi3 to upregulate expression during thiamine starvation [7]. We hypothesized that *CgPMU3* has a different sensitivity to Pdc2 relative to other THI promoters, either not requiring all of Pdc2, or being differentially sensitive to parts of Pdc2. To test this hypothesis, we asked if *CgPMU3* was more or less sensitive to fragmentation of Pdc2 -i.e. Does *CgPMU3* only need the activation domain of Pdc2 to express during thiamine starvation? We determined that both the DBD and the AD are required for expression of ancestral THI promoters and *CgPMU3*, and that there does not appear to be a difference between the promoters–all promoters require both the DBD and AD of Pdc2 (**Fig 7A**). We also observed that Pdc2 has a large, disordered region in the AD [29] (**S2 Fig**), and tested to see if *CgPMU3* was differentially sensitive to truncation of IDRs. The data indicate that there is not a large difference between how the THI promoters or *CgPMU3* respond to truncation of Pdc2 (**Fig 7B**). The data are consistent with the region having IDRs, as truncations lead to a gradual loss of transcriptional activity, similar to other studies [16], but further studies are warranted to determine if there really is a significant, differential requirement at *CgPMU3* relative to other THI promoters.

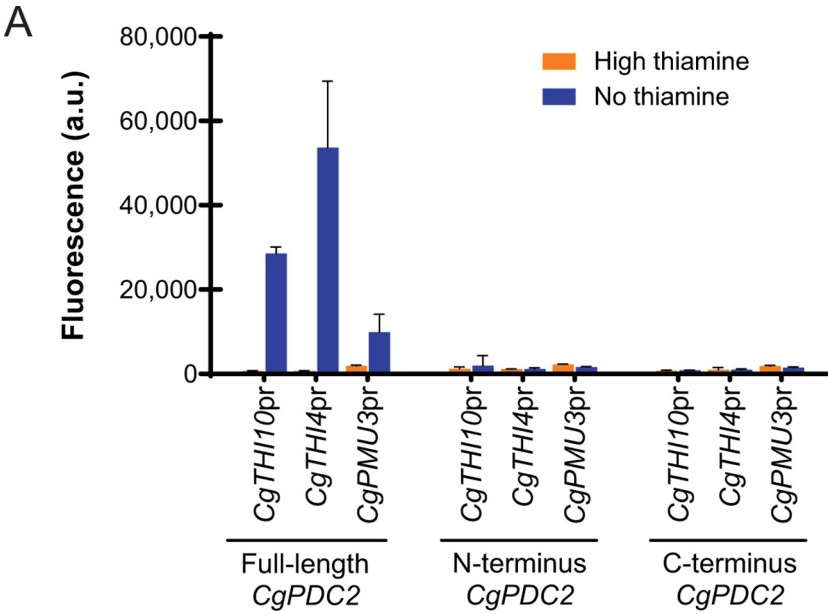

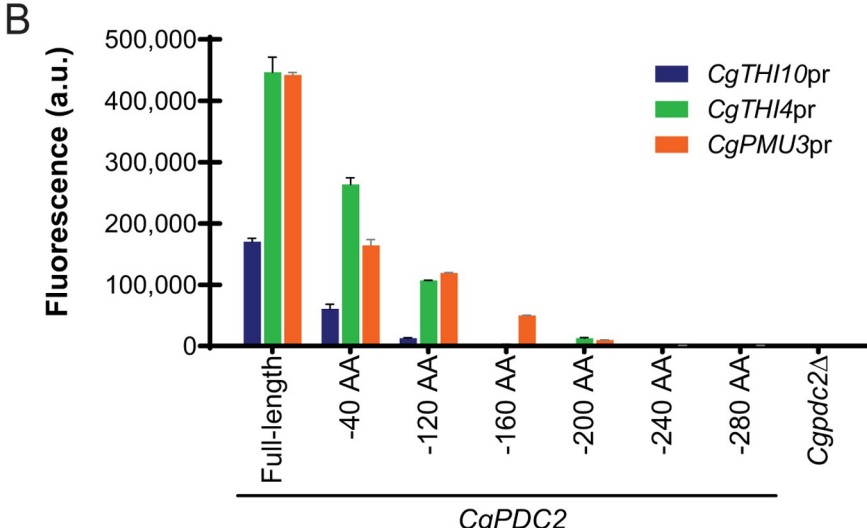

**Fig 7. Both the DNA binding domain (DBD) and activation domain (AD) of Pdc2 are required for the expression of all *C. glabrata* genes and sequential loss of the IDRs in the activation domain impacts *C. glabrata* promoters in the same way.** (A) Expression of *C. glabrata* promoters fused to YFP in a *Cgpdc2Δ* strain with *PDC2*, or just the DNA binding domain (DBD) or activation domain (AD) of *PDC2*, introduced on a plasmid. (B) Expression during thiamine starvation of *C. glabrata* promoters fused to YFP in a *Cgpdc2Δ* strain with *PDC2*, truncated from the C-terminal end of the activation domain, incorporated into the genome to minimize noise from plasmid expression. The data presented is the mean and standard deviation of three independently grown samples.

## Small sections of *C. glabrata* promoters can confer thiamine starvation regulation to an unregulated promoter

Given the multiple behaviors and dependencies of thiamine regulated promoters, we wanted to identify the smallest regions required to confer thiamine starvation regulation in three

promoters. We choose 1) the *ScPDC5* promoter because it appears to only be *PDC2* dependent (in both species), 2) the *CgTHI10* promoter because it requires both *THI3* and *PDC2* and does not function in *S. cerevisiae*, and 3) the *CgPMU3* promoter because it also requires *THI3* and *PDC2* but has different *cis* elements and does function in *S. cerevisiae*. To do this, we cloned 30 bp to 100 bp fragments around the *cis* elements [7] known to be important for regulation into the *CgPMU1* promoter (**Fig 8A**). We identified regions of varying lengths that are sufficient for conferring thiamine regulation to the *CgPMU1* promoter: 90 bp of the *ScPDC5* promoter, 60 bp of the *CgTHI20* promoter, and 100 bp of the *CgPMU3* promoter (**Fig 8B**). All of the promoter regions require *PDC2*, as deletion abrogates the upregulation during thiamine starvation, but *ScPDC5* continues to be *THI3* independent (**S3 Fig**). In the *ScPDC5* promoter, there are two 22 bp repeating elements that share similarity to the putative Pdc2 binding element identified previously [12], but *CgPMU3* does not appear to have a clear homologous Pdc2 binding sequence. Future studies will examine these sequences in detail.

## Discussion

Here, we observe that *Sc*Pdc2 and *Cg*Pdc2 have different roles in regulating genes. In *S. cerevisiae*, where fermentation is critical for much of its lifestyle, it is likely there is an advantage in having the same transcription factor be core to coupling thiamine biosynthesis and the major enzyme that uses thiamine, pyruvate decarboxylase. Addition of Thi2 likely allows for regulation of *Sc*Pdc2 to be further subdivided between THI and PDC genes. In *S. cerevisiae*, Pdc2 appears to work with Thi2 and Thi3 to regulate THI genes, and then separately, Pdc2 works alone (or in another complex) to drive transcription of the PDC genes.

In *C. glabrata*, Pdc2 appears to be core to just THI gene regulation. Thi3 and Pdc2 work together to drive transcription, and the C-terminal domain of Pdc2 is important for that complex to function. This is supported by *C. glabrata* THI genes being similarly sensitive to loss of both *CgPDC2* and *CgTHI3*. However, the PDC genes appear to be regulated by transcription factors other than Pdc2, although the cross species complementation experiments seem to suggest that the *cis* elements required for Pdc2 regulation are still present. The presence of potential Pdc2 regulation suggests there is cryptic dual regulation of PDC and THI genes in *C. glabrata* that we are not able to observe in our experimental set up. It is possible that Pdc2 regulates PDC genes in *C. glabrata* but only in another growth condition that we did not examine in our studies.

The *CgPMU3* promoter is recently evolved, does not have clear *cis* elements in common with other THI promoters in either species, and is Pdc2- and Thi3-dependent, raising the question of how this promoter functions. Because promoters between the two species are different enough from one another that they cannot be aligned, it is difficult to identify individual precise *cis* elements. However, having promoters from both species and a novel, newly evolved promoter narrowed down to ~100 bp now provides a lot of raw material for the eventual identification of sequences that are maintained for appropriate regulation. Future studies should be able to identify precise Pdc2 binding sites and those should correlate to conserved regions of each promoter. It will be interesting to determine whether the *CgPMU3* promoter has acquired a cryptic Pdc2 binding site, or if a different *cis* element has recruited Pdc2 in a different fashion as other Pdc2 dependent promoters. One possibility is that we have not examined large enough promoter sequences, but typically promoters in these yeast species are small and we have demonstrated for some THI and PDC genes that examining 1 or 2 kb of the promoter recapitulates the same function (**S4 Fig**).

Interestingly, *Sc*Thi3 has been posited to be the thiamine sensor (or more precisely the TPP sensor), but in both species *ScPDC5* is capable of being highly induced during thiamine

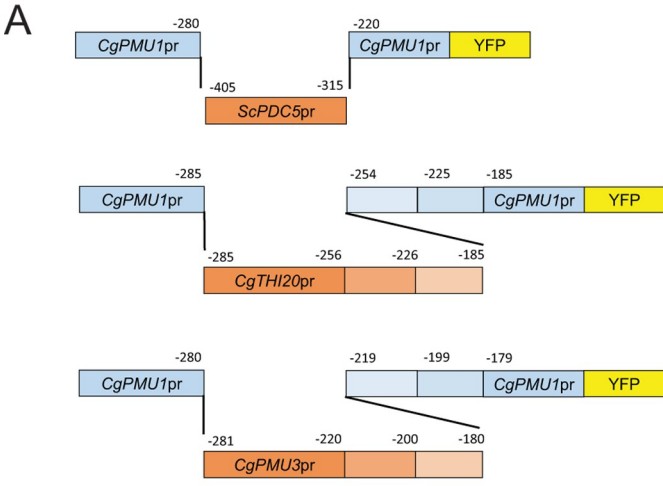

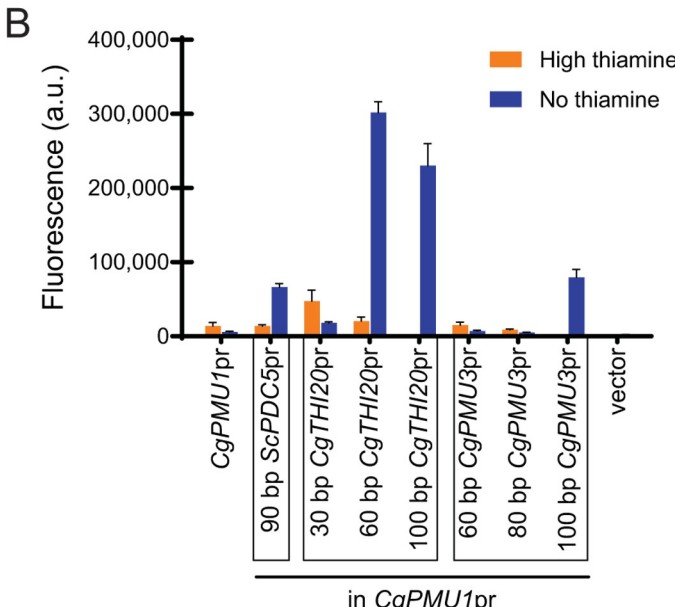

**Fig 8. Short regions of *C. glabrata* THI promoters, or *ScPDC5* promoter, can confer thiamine starvation regulation to an unregulated promoter.** Regions from *ScPDC5*, *CgTHI20*, and *CgPMU3* promoters were incorporated into the *CgPMU1* promoter, which is not regulated by thiamine, fused to YFP. (A) Schematic demonstrating construction of the *CgPMU1* promoter with regions from the *ScPDC5* and *C. glabrata* THI promoters incorporated. (B) Expression was measured in high thiamine and thiamine starvation (no thiamine). The data presented is the mean and standard deviation of three independently grown samples.

starvation independent of Thi3. This suggests that Pdc2 or another unknown factor is capable of sensing thiamine concentrations. However, an alternative hypothesis is possible. Upregulation of *ScPDC5* could be an indirect consequence of thiamine starvation. *Sc*Pdc1 protein could have a lower affinity for TPP than *Sc*Pdc5, and there could be feedback mechanisms leading to the increase in *ScPDC5* expression during thiamine starvation because of pyruvate (or another metabolite) accumulating and activating *Sc*Pdc2 binding at the *ScPDC5* promoter. Further

studies are required to tease apart the regulation of PDC genes and answer the questions of when Pdc2 is binding and what complexes are binding at PDC promoters.

This work supports the argument that Pdc2 participates in multiple complexes to drive transcription of different classes of genes. The canonical complex has been Thi2/Thi3/Pdc2; however, we present evidence that loss of Thi3 actually enhances the expression of most PDC genes, indicating that the PDC genes bind Pdc2 as a different complex. Additionally, the *ScPDC5* promoter suggests that cells sense thiamine concentration independent of Thi3. Our cloning of small *cis* regions sufficient to confer regulation should help us to understand what *trans* elements bind where. These *cis* elements require a complicated *trans* environment that we still do not understand. Future studies will identify the critical elements in these small regions and determine where transcription factors bind using chromatin immunoprecipitation.

## Supporting information

**S1 Table. Strains used in this paper.**
(XLSX)

**S2 Table. Primers used in this paper.**
(XLSX)

**S1 Fig. Loss of *PDC1* leads to an upregulation of *PDC5* in *S. cerevisiae* in both high thiamine and thiamine starvation conditions.** Expression of *ScPDC5* promoter fused to YFP in high thiamine and thiamine starvation (no thiamine) in strains where the thiamine regulatory factors and *PDC1* were deleted in *S. cerevisiae*. The data presented is the mean and standard deviation of three independently grown samples.
(EPS)

**S2 Fig. The activation domains of *C. glabrata* and *S. cerevisiae* Pdc2 proteins are made up of intrinsically disordered regions.** Graph showing the predicted intrinsically disordered regions of the *C. glabrata* and *S. cerevisiae* Pdc2 proteins, where higher values correspond to a higher probability of disorder. Disorder tendency was calculated using the online tool IUPred2.
(EPS)

**S3 Fig. The regions from *PDC5* and THI promoters that confer regulation by thiamine to the *CgPMU1* promoter are dependent on Pdc2.** Regions from *ScPDC5*, *CgTHI20*, and *CgPMU3* promoters were incorporated into the *CgPMU1* promoter fused to YFP, allowing this promoter to be upregulated during thiamine starvation. These promoter regions require Pdc2 and Thi3, as deletion abrogates expression, with the exception of *ScPDC5*, which remains Thi3 independent. The data presented is the mean and standard deviation of three independently grown samples.
(EPS)

**S4 Fig. THI and PDC promoters in *S. cerevisiae* and *C. glabrata* are appropriately regulated by thiamine starvation regardless of the length of the promoter.** To determine if varying lengths of promoter sequence change the expression of THI genes, 2 Kb or 1 Kb upstream of the start codon was fused to YFP in a wild-type strain and expression of these promoters was measured in thiamine starvation. Expression of these promoters was fully repressed during high thiamine conditions, with the exception of the *CgPDC1* promoter, which is not regulated by thiamine starvation. The data presented is the mean and standard deviation of three independently grown samples. (A) *S. cerevisiae* promoters in *S. cerevisiae* wild-type. (B) *C.*

*glabrata* promoters in *C. glabrata* wild-type.
(EPS)

## Author Contributions

**Conceptualization:** Christine L. Iosue, Dennis D. Wykoff.

**Data curation:** Julia M. Ugras, Cory A. Dottor, Peyton L. Stauffer, Rachael A. Hopkins, Emma C. Lang, Dennis D. Wykoff.

**Formal analysis:** Christine L. Iosue, Julia M. Ugras, Yakendra Bajgain, Cory A. Dottor, Rachael A. Hopkins, Emma C. Lang, Dennis D. Wykoff.

**Funding acquisition:** Dennis D. Wykoff.

**Investigation:** Christine L. Iosue, Julia M. Ugras, Yakendra Bajgain, Cory A. Dottor, Peyton L. Stauffer, Rachael A. Hopkins, Emma C. Lang, Dennis D. Wykoff.

**Project administration:** Dennis D. Wykoff.

**Supervision:** Christine L. Iosue.

**Validation:** Yakendra Bajgain, Cory A. Dottor, Peyton L. Stauffer, Rachael A. Hopkins, Emma C. Lang, Dennis D. Wykoff.

**Writing – original draft:** Christine L. Iosue, Julia M. Ugras, Yakendra Bajgain, Emma C. Lang, Dennis D. Wykoff.

**Writing – review & editing:** Christine L. Iosue, Cory A. Dottor, Peyton L. Stauffer, Rachael A. Hopkins, Dennis D. Wykoff.

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
