## [Decision Letter · Decision Letter 0]

4 Apr 2023

PONE-D-23-06178Pyruvate decarboxylase and thiamine biosynthetic genes are regulated differently by Pdc2 in S. cerevisiae and C. glabrata.PLOS ONE

Dear Dr. Wykoff,

Thank you for submitting your manuscript to PLOS ONE. After careful consideration, we feel that it has merit but does not fully meet PLOS ONE’s publication criteria as it currently stands. Therefore, we invite you to submit a revised version of the manuscript that addresses the points raised during the review process. Please submit your revised manuscript by May 19 2023 11:59PM. If you will need more time than this to complete your revisions, please reply to this message or contact the journal office at plosone@plos.org. Please include the following items when submitting your revised manuscript:A rebuttal letter that responds to each point raised by the academic editor and reviewer(s). You should upload this letter as a separate file labeled 'Response to Reviewers'.A marked-up copy of your manuscript that highlights changes made to the original version. You should upload this as a separate file labeled 'Revised Manuscript with Track Changes'.An unmarked version of your revised paper without tracked changes. You should upload this as a separate file labeled 'Manuscript'.If applicable, we recommend that you deposit your laboratory protocols in protocols.io to enhance the reproducibility of your results. Protocols.io assigns your protocol its own identifier (DOI) so that it can be cited independently in the future. For instructions see: https://journals.plos.org/plosone/s/submission-guidelines#loc-laboratory-protocols. Additionally, PLOS ONE offers an option for publishing peer-reviewed Lab Protocol articles, which describe protocols hosted on protocols.io. Read more information on sharing protocols at https://plos.org/protocols?utm_medium=editorial-email&utm_source=authorletters&utm_campaign=protocols.

We look forward to receiving your revised manuscript.

Kind regards,

Hari S. Misra

Academic Editor

PLOS ONE

Journal Requirements:

Reviewers' comments:

Reviewer's Responses to Questions

**Comments to the Author**

1. Is the manuscript technically sound, and do the data support the conclusions?

Reviewer #1: Yes

Reviewer #2: Yes

2. Has the statistical analysis been performed appropriately and rigorously? 

Reviewer #1: Yes

Reviewer #2: Yes

3. Have the authors made all data underlying the findings in their manuscript fully available?

Reviewer #1: No

Reviewer #2: Yes

4. Is the manuscript presented in an intelligible fashion and written in standard English?

Reviewer #1: Yes

Reviewer #2: Yes

5. Review Comments to the Author

Reviewer #1: This is a nice article about differences between S. cerevisiae and C. glabrata thiamine biosynthesis genes, their promoters and regulation. It is written well and the data is shown in a clear and simple manner. The results show that S. cerevisiae and C. glabrata differ in their ability to respond to thiamine deprivation, including the promoter responses of key thiamine-inducible genes. There are differences in the genomes of the two organisms, and the Thi2 transcriptional regulator is missing in C. glabrata.

While these results are perhaps not surprising, the authors have carried out a detailed analysis of the transcriptional responses of different promoters in different knock-out strains of S. cerevisiae and C. glabrata and show that regulation is different on the two organisms.

Major comments:

1) The major concern is that the analysis of the promoters is done uniformly with 1 kb of sequence upstream of the start codon. Is this sequence inclusive of all promoter elements? Has it been characterized? Is the sequence similar in S. cerevisiae and C. glabrata? If the sequence is very different in the two organisms, then the comparisons are not meaningful. It would be really helpful to get information about the 1kb sequences of the promoters from both S. cerevisiae and C. glabrata, in terms of their sequence identity.

2) The authors mention that the CgPMU3 promoter is different. It is not clear how. This needs to be explained and the points made above are relevant to this as well.

3) In Figure 4, S. cerevisiae promoters are being tested in C. glabrata and and vice versa. Again, if the sequences are very different, these results simply show that the promoters are not the same, rather than differences in regulation. The bottom line is that it would be very important to show some data about the sequences of the 1 kb regions and compare them.

Minor comments:

1) At the end of the paper, the reader is left with the main finding that the thiamine-inducible gene promoters are different in S. cerevisiae and C. glabrata. It would be very helpful to have a schematic to bring all the information together.

2) The authors talk about “complexity” of regulation (e.g. Abstract, lines 35,254, 279, 281, 336, 339, 364). What do they mean by this word? The results show differences but it is not clear why the words “complexity/complex” is used to describe these differences

Reviewer #2: In this manuscript, the authors have addressed the differential regulation of Pyruvate decarboxylase and thiamine biosynthetic genes by the regulator Pdc2 in two yeast species S. cerevisiae and C. glabrata. The differences are addressed at the level of differences in the cis-acting components in the promoters and the structural organization of the trans-acting components of the regulator protein Pdc2 in the two species.

Although the two species differ in that deletion of PDC2 is lethal in S. cerevisiae but not in C. glabrata, the cross-complementation studies reveal presence of cryptic cis elements in the C. glabrata promoters that are not apparently regulated by CgPdc2 but can be regulated by the ScPdc2. Among other difference could be ascribed to absence of Thi2 in C. glabrata, with its inclusion in S. cerevisiae possibly playing a role in imparting greater complexity of regulation of THI and PDC genes through the formation of a compendium of transcription regulatory Pdc2-containing complexes S. cerevisiae but a simpler regulatory machinery independent of Pdc2 in C. glabrata. The authors also demonstrate the role of Intrinsically disordered regions in the activation function of the C-terminal region of CgPdc2.

The authors opine that a deeper analysis of the different regulation among the two species could help identify potential druggable targets that are unique to C. glabrata.

I find this study well-structured and conclusions consistent with the results. Accordingly, I recommend it for publication in PlosOne.

However, I think that the manuscript could improve in clarity and continuity of logic, especially in the discussion section, where some sentences are lacking in clarity.

For example: lines 425-430 , 434-436.

6. PLOS authors have the option to publish the peer review history of their article (what does this mean?). If published, this will include your full peer review and any attached files.

Reviewer #1: No

Reviewer #2: No

---

## [Author Response · Author response to Decision Letter 0]

16 May 2023

Response to reviewers

Reviewer #1: This is a nice article about differences between S. cerevisiae and C. glabrata thiamine biosynthesis genes, their promoters and regulation. It is written well and the data is shown in a clear and simple manner. The results show that S. cerevisiae and C. glabrata differ in their ability to respond to thiamine deprivation, including the promoter responses of key thiamine-inducible genes. There are differences in the genomes of the two organisms, and the Thi2 transcriptional regulator is missing in C. glabrata.

While these results are perhaps not surprising, the authors have carried out a detailed analysis of the transcriptional responses of different promoters in different knock-out strains of S. cerevisiae and C. glabrata and show that regulation is different on the two organisms.

Major comments:

1) The major concern is that the analysis of the promoters is done uniformly with 1 kb of sequence upstream of the start codon. Is this sequence inclusive of all promoter elements? Has it been characterized? Is the sequence similar in S. cerevisiae and C. glabrata? If the sequence is very different in the two organisms, then the comparisons are not meaningful. It would be really helpful to get information about the 1kb sequences of the promoters from both S. cerevisiae and C. glabrata, in terms of their sequence identity.

We have now included an additional supplementary figure (S4 Fig) indicating that whether we choose 1kb or 2 kb upstream of the start codon, we observe similar expression for six of the promoters that we examine. It is possible that indeed we are missing some subtleties of expression by only examining 1 kb promoters, but we do believe we are capturing most of the expression regulation in the 1 kb promoter. In our experience in analyzing promoters, we often find that 500 bp recapitulates much of a promoter response, and core promoter elements (TATA boxes, etc.) are often 100-150 bp upstream of the start codon. There are clearly exceptions, but we believe our supplementary figure should assuage these concerns.

Alignment of orthologous promoters from the two species do not result in meaningful alignments. Using relaxed BLASTn algorithms can sometimes find significant similarity in the promoters, but overall there is too much sequence divergence. At the protein level, the two species share 60-80% identity, but at the DNA level, similarity drops dramatically. These two species diverged from one another too long ago for promoters to be able to be aligned. However, we would argue that comparing the promoters is useful. We observe similar cis elements in both species’ promoters, and the ScPDC5 promoter contains cis elements that are recognized by Pdc2 in each species. Our study was designed to begin to understand how similar the promoter architecture is between these two yeast species that likely diverged from one another millions of generations before present. We believe it underlies that these promoter behaviors have been strongly selected for given little sequence similarity, except at likely Pdc2 cis elements.

2) The authors mention that the CgPMU3 promoter is different. It is not clear how. This needs to be explained and the points made above are relevant to this as well.

We agree with the reviewer that we were not clear about why we think this promoter is interesting. We have inserted some clarification and background. We find it exciting that C. glabrata has evolved a brand-new phosphatase gene (CgPMU3) that cleaves thiamine pyrophosphate (TPP), is required for C. glabrata to survive when TPP is the only thiamine source, and that this promoter acquired the ability to be regulated by the same transcription factors as other THI genes. Because CgPMU3 promoter does not use a cis element that is common to other THI promoters, it begins to give us clues as to how a new promoter can evolve to be regulated simply by promoter substitutions. We think in the long-term, this promoter will give us clues as to how the transcriptional complexes function at THI promoters, but we do not have enough information yet.

3) In Figure 4, S. cerevisiae promoters are being tested in C. glabrata and and vice versa. Again, if the sequences are very different, these results simply show that the promoters are not the same, rather than differences in regulation. The bottom line is that it would be very important to show some data about the sequences of the 1 kb regions and compare them.

We addressed this important concern in the answer above. We disagree with the reviewer that the data simply show the promoters are not the same. We believe what is key here is that many of the promoters between species have similar patterns of expression (e.g. regulation by thiamine and different regulation of the promoter when THI3 is deleted). It is a “feature, rather than a bug” of our study that the promoter sequence between the two species is very different. Thus, when we know where transcription factors bind, we can identify the key cis elements that dictate this behavior. We have promising preliminary results that suggest that CgPdc2 binds to specific elements using ChIP-seq, but we believe that data is best in another publication.

Minor comments:

1) At the end of the paper, the reader is left with the main finding that the thiamine-inducible gene promoters are different in S. cerevisiae and C. glabrata. It would be very helpful to have a schematic to bring all the information together.

We agree with the reviewer. We spent a lot of time trying to figure out how to condense the data that we have generated into a schematic, and we found all of them wanting. We believe that the reviewer may be indicating the same issues in the next concern and mirroring the concerns of reviewer #2 that there is a weakness in the “clarity and logic” of the discussion and we have worked to make the discussion clearer such that a schematic is not needed. Unfortunately, we are still trying to figure out how everything works together and so there is going to be some lack of clarity in our discussion. We have endeavored to make the writing clear and to highlight the outstanding questions that we still have.

2) The authors talk about “complexity” of regulation (e.g. Abstract, lines 35,254, 279, 281, 336, 339, 364). What do they mean by this word? The results show differences but it is not clear why the words “complexity/complex” is used to describe these differences

We agree with the reviewer that we were not clear about what we mean. We have edited the manuscript to avoid these vague statements and appreciate both reviewers concerns that there is a vagueness in parts of the manuscript.

Reviewer #2: In this manuscript, the authors have addressed the differential regulation of Pyruvate decarboxylase and thiamine biosynthetic genes by the regulator Pdc2 in two yeast species S. cerevisiae and C. glabrata. The differences are addressed at the level of differences in the cis-acting components in the promoters and the structural organization of the trans-acting components of the regulator protein Pdc2 in the two species.

Although the two species differ in that deletion of PDC2 is lethal in S. cerevisiae but not in C. glabrata, the cross-complementation studies reveal presence of cryptic cis elements in the C. glabrata promoters that are not apparently regulated by CgPdc2 but can be regulated by the ScPdc2. Among other difference could be ascribed to absence of Thi2 in C. glabrata, with its inclusion in S. cerevisiae possibly playing a role in imparting greater complexity of regulation of THI and PDC genes through the formation of a compendium of transcription regulatory Pdc2-containing complexes S. cerevisiae but a simpler regulatory machinery independent of Pdc2 in C. glabrata. The authors also demonstrate the role of Intrinsically disordered regions in the activation function of the C-terminal region of CgPdc2.

The authors opine that a deeper analysis of the different regulation among the two species could help identify potential druggable targets that are unique to C. glabrata.

I find this study well-structured and conclusions consistent with the results. Accordingly, I recommend it for publication in PlosOne.

However, I think that the manuscript could improve in clarity and continuity of logic, especially in the discussion section, where some sentences are lacking in clarity.

For example: lines 425-430 , 434-436.

We appreciate the reviewer’s comments, and agree that there are some lapses in our logic. We have reworked the discussion and clarified our use of “complexity” throughout the manuscript.

---

## [Editor Report · Decision Letter 1]

23 May 2023

Pyruvate decarboxylase and thiamine biosynthetic genes are regulated differently by Pdc2 in S. cerevisiae and C. glabrata.

PONE-D-23-06178R1

Dear Dr. Wykoff,

We’re pleased to inform you that your manuscript has been judged scientifically suitable for publication and will be formally accepted for publication once it meets all outstanding technical requirements.

Kind regards,

Hari S. Misra, Ph.D.

Academic Editor

PLOS ONE
---

## [Editor Report · Acceptance letter]

29 May 2023

PONE-D-23-06178R1 

Pyruvate decarboxylase and thiamine biosynthetic genes are regulated differently by Pdc2 in *S. cerevisiae* and *C. glabrata*

Dear Dr. Wykoff:

I'm pleased to inform you that your manuscript has been deemed suitable for publication in PLOS ONE. Congratulations! Your manuscript is now with our production department. 

Kind regards, 

on behalf of

Professor Hari S. Misra 

Academic Editor

PLOS ONE